# Prostaglandins: Biological Action, Therapeutic Aspects, and Pathophysiology of Autism Spectrum Disorders

**DOI:** 10.3390/cimb47020071

**Published:** 2025-01-21

**Authors:** Kunio Yui, George Imataka, Mariko Ichihashi

**Affiliations:** 1Department of Pediatrics, Chiba University, Chiba-Shi 260-8677, Chiba, Japan; 2Department of Pediatrics, Dokkyo Medical University, Tochigi 321-0293, Tochigi, Japan; geo@dokkyomed.ac.jp; 3Department of Orthopedic Surgery, Hyogo Medical University, Nishinomiya 663-8501, Hyogo, Japan

**Keywords:** prostaglandins, autism spectrum disorders, prostaglandin E2 (PGE2), cyclooxygenase (COX2), COX2/PEG2 signaling

## Abstract

Esterified ARA on the inner surface of the cell membrane is hydrolyzed to its free form by phospholipase A2 (PLA2), which is further metabolized by COXs and lipoxygenases (LOXs) and cytochrome P450 (CYP) enzymes. PGs produce detrimental effects due to their proinflammatory properties. The generation of prostaglandin (PG)G_2_ and PGH_2_ is triggered by cyclooxygenase (COX) isozymes such as COX-1 and COX-2. Prostaglandin E2 (PGE2) is significantly elevated in ASD. Considerable data indicate that COX enzymes and their metabolites of ARA play important roles in the initiation and development of human neurodevelopmental diseases. The involvement of disrupted COX2/PGE2 signaling in ASD pathology in changing neuronal cell behavior and the expression of ASD-related genes and proteins is due to disrupted COX2/PGE2 signaling. Prostacyclin (PGI2) is synthesized from arachidonic acid by metabolic-pathway-dependent cyclooxygenase (COX) and synthesized in a primary step of ARA transformation (PGG2, PGH2), by degradation of the abovementioned prostaglandins.

## 1. Introduction

Phospholipase A2 (PLA2) plays a fundamental role in the production of prostaglandins (PGs) by releasing arachidonic acid (ARA) [1]. Esterified ARA on the inner surface of the cell membrane is hydrolyzed to its free form by PLA2. PGs are synthesized from ARA by the action of cyclooxygenases (COXs) and terminal PG synthases [2]. Cellular and mitochondrial membrane phospholipids provide the synthesis and release of PGs in response to chemical and noxious stimuli [2]. Abnormalities in the COX-2/prostaglandin E2 (PGE2) pathway due to genetic or environmental factors have been linked to autism spectrum disorders (ASDs) [3].

Deficient dietary supplementation and exposure to oxidative stress, infections, and inflammation may disrupt signaling of the PGE_2_ pathway and contribute to ASD. The structure and establishment of two critical protective barriers in the brain during early development are influenced by environmental factors, such as exposure to chemicals in air pollution, pesticides, and consumer products. These factors may disrupt PGE2 signaling, thereby increasing the risk of developing ASD. Additionally, it is crucial to understand how these exogenous agents are capable of crossing protective barriers, highlighting the importance of avoiding or limiting exposure to these factors during vulnerable periods of development [4].

## 2. Metabolites of Prostaglandins (PGs)

Prostaglandin E2 (PGE_2_) is an endogenous lipid mediator of inflammation. Its production is regulated by the rate-limiting upstream enzyme cyclooxygenase-2 (COX-2) [5], regulating membrane excitability in various forms of synaptic plasticity. Spontaneous glutamatergic synaptic activity regulates constitutive neuronal COX-2 expression [6].

Esterified ARA on the inner surface of the cell membrane is metabolized to a free form by PLA2 and further metabolized by COXs, lipoxygenases (LOXs), and cytochrome P450 (CYP) enzymes into a spectrum of bioactive mediators (Figure 1). The COX enzymes and ARA metabolites play important roles in the initiation and development of human diseases [7]. The generation of PGG2 and PGH2 is initiated by COX isozymes, including COX-1 and COX-2, which are derived from ARA through the enzymatic activities of COX and peroxidase. [8]. ARA is metabolized to eicosanoids, including PGs, leukotrienes, and thromboxanes, which are proinflammatory molecules, triggered by oxidative stress [9]. These lipid mediators play critical roles in the initiation, maintenance, and modulation of neuroinflammation and oxidative stress [9]. Therefore, the ARA metabolic pathways are effective targets for managing inflammatory-related diseases [10]. Eicosanoids play a role in neural functioning, including sleep induction (PGD2), long-term potentiation, spatial learning, and synaptic plasticity (PGE2) [11].

## 3. Role of Prostaglandins

PGF2a and thromboxane A2 (TXA2) are endogenous ARA metabolites, influencing diverse physiological processes such as inflammation via activation of PGF2a receptor (FP) and TXA2 receptor (TP) [12].

Endogenous and exogenous PGs have beneficial effects on cellular protection, sleep regulation, and blood flow enhancement [2]. PGs and their receptors are intimately involved in many homeostatic activities and are beneficial in the proper functioning of the central nervous system (CNS), as they are the structural basis for ligand recognition and the activation of prostanoid receptors [2]. The various roles of PGs indicate destructive or neuroprotective features of these ubiquitous endogenous mediators [2].

PGE2 and PGD2, which are biosynthesized from ARA by enzymatic cleavage of membrane phospholipids in response to various stimuli, play key roles in multiple brain pathophysiological processes, including modulation of synaptic plasticity, neuroinflammation, and sleep promotion [13]. Prostacyclin 12 (PGI2) is synthesized from ARA by metabolic-pathway-dependent COX [14]. PGD2 is synthesized in the primary step of ARA transformation (PGG2, PGH2) by degradation of the abovementioned PGs but is not presented physiologically, and may be chemically generated (PGK2, PGL2) [14]. PGG_2_ and PGH_2_ are produced from ARA, which act as inflammatory cells [8].

Regarding the association among PUFAs and long-chain metabolites, including gamma-linolenic acid (GLA), dihomo-GLA (DGLA), ARA, eicosapentaenoic acid (EPA), and DHA, other PG products (PGE1 and PGI2) and lipoxins (LXs) may have anti-inflammatory actions [15].

## 4. PEG2 and Glutamate Activity

PGE2 inhibits the neuronal activity of the dorsolateral periaqueductal gray via attenuation of glutamatergic synaptic inputs [16]. Glutamate excitotoxicity was proposed to participate in the propagation of ASD [17]. Palmitoylethanolamide (PEA) is an endocannabinoid proven to prevent glutamatergic toxicity and inhibit inflammatory responses simultaneously [17]. PGE2 is a physiological signal for inducing glutamate release from subventricular zone in astrocytes, thus being important for moderating neuroblast survival and proliferation [17].

Glutamate plays an important role in brain development, neuronamigration, differentiation, survival and synaptogenesis, resulting in good a potential biomarker of ASD [18]. Moreover, the genetic excitatory/inhibitory imbalance between excitatory (glutamatergic) and inhibitory (GABAergic) mechanisms underlies autistic behaviors [19]. Therefore, PEG 2E may be closely related to glutamate toxication, contributing to the development of ASD [18].

Glutamate excitotoxicity was proposed to participate in the propagation of ASD [17]. Abnormalities in glutamate and GABA signaling have been hypothesized to underlie ASD symptoms and may form a therapeutic target in humans with ASD and animal models of the disorder. In humans with ASD, glutamate concentration was reduced in the striatum in correlation with the severity of social symptoms. GABA levels were not altered in either brain region [18]. Glutamate/GABA abnormalities in the corticostriatal circuitry may be a key pathological mechanism in ASD and may be linked to alterations in the neuroligin–neurexin signaling complex. Adults with idiopathic ASD have lower glutamate concentrations in the striatum compared to controls, suggesting a correlation with the severity of social impairment. [18]. Brain region- and even neural circuit-specific changes in the E/I balance have also been observed in animal models of ASD. These rodent models of ASD are similar to each other with regard to this key neurochemical change, and they recapitulate the change seen in human idiopathic ASD [18]. A correlation between striatal glutamate levels and the severity of social deficits has been suggested [18].

The decrease in striatal glutamate is related to core ASD symptom severity. Glutamatergic dysfunction in the corticostriatal pathway is an underlying core pathophysiological mechanism of ASD. The findings on humans and rodent models of ASD are suggestive of a region-specific imbalance slanted toward reduced excitation [18].

In summary, blood glutamate levels might be a potential biomarker of ASD [19]. Additionally, imbalance between excitatory (glutamatergic) and inhibitory (GABAergic) mechanisms underlies the behavioral characteristics of ASD [20].

## 5. Association of Prostaglandins with Other Amino Acids

GABAergic inhibitory neurotransmission is critical for the regulation of brain rhythm and spontaneous neuronal activities during neurodevelopment via an abnormal excitatory/inhibitory (E/I) neurotransmission ratio implicated in the pathogenesis of ASD. Current candidate genes induce dysfunction of GABAergic transmission by affecting the transcription of GABA-A receptor (GABA-AR) and causing presynaptic GABA release, the formation of GABAergic synapses, and synaptic structure-mediated transmission [21].

Impaired GABAergic signaling during brain development is a crucial event implicated in the pathogenesis of neurodevelopmental diseases. In addition, 15q duplication mice (a model for ASD with human chromosome 15q11–q13 paternal duplication) exhibited facilitated activity-induced long-term potentiation (LTP) of glutamate synapses onto layer-5 pyramidal neurons and a reduced number of inhibitory synapses, indicating that shifted E/I balance is implicated in autism [21].

GABAergic dysfunction could be a downstream consequence of mutations in genes and not directly be involved in GABA transmission because the proper development and function of GABA synapses rely on numerous signaling and scaffolding proteins. Currently known candidate genes of ASD include transcription factors, scaffolding proteins, receptors, and signaling pathways. GABA functions as an excitatory neurotransmitter during embryonic development and may play a role in the pathophysiology of fragile X syndrome. ASD models have an abnormal excitatory GABA inhibitory shift during delivery with elevated intracellular chloride levels and maintenance of higher levels of excitatory GABA, promoting glutamic acid activity and gamma oscillatory activity [21].

The transition of inhibitory to excitatory neurotransmission during early brain development might be s pathological mechanism of the imbalance in the E/I ratio and GABA disturbance involved in ASD. Specific brain regions and developmental time windows play a critical role in understanding the mechanisms of GABA involved in ASD [21].

Valproic acid (VPA), as a broad-spectrum antiepileptic drug, has a significant facilitative effect on the GABAergic neurotransmission system. Impaired GABAergic inhibitory transmission has been observed in prenatal VPA exposure-induced autism models. Prenatal maternal immune activation might have effect on long-lasting GABAergic changes relevant to ASD.

In summary, duplications of the E3 ubiquitin ligase gene UBE3A and three non-imprinted GABA-AR genes located at 15q11.2–q13.1 are associated with autistic phenotype behavioral deficits. This is linked to the structural chromosomal protein MeCP2, a homeobox transcription factor involved in the synthesis and modification of synaptic proteins crucial for GABAergic synaptic function. Neuroligins might affect the formation of neural connectivity via affecting the early postnatal transition of excitatory GABA. Pharmacological intervention with agonists targeting specific GABA receptors alleviates ASD-related behavioral abnormalities in animal models [21].

GABAergic synaptic transmission dysfunction in specific neurons or circuitry during the developmental stage might underlie the pathophysiology of ASD. Impaired GABAergic transmission might be a critical etiology of imbalance of the E/I ratio involved in the development of ASD. GABA can shift from “excitatory” to “inhibitory” and play a critical role in the subtle regulation of neuronal networks during critical developmental stages, and altered excitation might be secondary to reduced inhibition and GABA dysfunction [21].

PGE_2_ potently depresses GABAergic-inhibitory synaptic transmission on parvocellular neuroendocrine cells (PNCs). Using pharmacological approaches via the actions of presynaptic EP3 receptors potently depresses GABA release onto parvocellular neuroendocrine cells (PNCs) [22].

PGE2 acts at a presynaptic locus to decrease the probability of GABA release. The PEG receptor EP3 subtype of the PGE2 receptor mediates the actions of PGE2 on GABA synapses. PGE2, via actions of presynaptic EP3 receptors, potently depresses GABA release onto PNCs, providing a plausible mechanism for the disinhibition of HPA axis output during inflammation. The effects of PGE2 on GABA synapses provide strong evidence indicating that PGE2 potently inhibits GABAergic synaptic transmission (eIPSCs, sIPSCs and mIPSCs. The PGE2-induced GABA synapse depression is primarily attributable to a decrease in the release of vesicles from presynaptic terminals. PGE2-attenuated GABA release is independent of the potential firing of presynaptic neurons and its receptors expressed on the axonal terminals in the paraventricular hypothalamic nucleus [23].

## 6. Prostaglandin Signaling in Pathophysiology of ASD

### 6.1. PEG2-COX2 I ASD

The presence of an abnormal COX-PGE_2_ signaling pathway due to genetic or environmental causes may be linked to ASD [23]. The brain at embryonic day 16 (E16) is particularly susceptible to the influence of biological pathways involving various ASD-associated genes. Crosstalk between the COX2–PGE2 and Wnt pathways during early brain development may play a crucial role in the pathophysiology of ASD models [23].

Higher levels of PGE2 increased the Wnt-dependent motility and proliferation of neuroectodermal stem cells and modified the expression of Wnt genes linked to ASD [23]. PGE2 signaling also converges with the Wnt canonical pathway in the developing mouse brain after maternal exposure to PGE2 at the onset of neurogenesis [23]. Since prenatal exposure to PGE2 affects the Wnt pathway at the level of β-catenin, the major downstream regulator of Wnt may be related to gene transcription [23]. PGE2 can influence the expression level of genes from the canonical Wnt signaling pathway, including Mmp7, Wnt2, and Wnt3a, linked to ASD [23]. The convergence of PGE2 with the Wnt canonical pathway was apparent due to higher levels of PKA-activated phosphorylated β-catenin (Ser552) and active non-phospho β-catenin (Ser33/37/Thr41) [23] (Figure 2).

### 6.2. Mechanisms of PEG2

Importantly, changes in the matrilysin activity level, also known as matrix metalloproteinase-7 (Mmp7), due to exposure to PGE2 may have potential consequences at various stages of the developing brain. Abnormal PGE2 levels in the mouse brain during the developmental period may interfere with the Wnt pathway via phosphorylation of β-catenin at multiple sites, which can lead to differential expression of crucial neurodevelopmental genes [23]. Exposure to maternal PGE2 at a critical time can have an effect on the expression of Wnt target genes during prenatal (E16 and E19) brain development [23].

## 7. PGs in the Pathophysiology of ASD

PGE2 is an endogenous lipid molecule that plays an important role in normal brain development. COX2 is the primary regulator of PGE2 production [22]. In a study by El-Ansary, A, the significant decrease in their ratio (COX2/PGE2) as a marker of lipid metabolism signaling in autistic individuals compared to controls may be due to altered COX-2 enzyme kinetics in ASD patients. PGE2 regulates the effects of Wnt signaling through cAMP/PKA activity, and it may directly regulate β-catenin destruction by the nucleosome and subsequent proteins available for transcriptional activation [22]. ASD patients with an abnormal COX-2/PGE2 signaling pathway have a significantly lower ratio, despite higher levels of independent variables. The combined ROC of COX/PGE2 and anti-nucleosome autoantibodies, along with their proven roles in impaired lipid metabolism, autoimmunity, and glutamate excitotoxicity as the disorder’s three etiological mechanisms, raises the possibility that these variables could be used for an early diagnosis of ASD and certain co-morbidities seen in ASD patients [22].

PGE2 interacts with canonical Wnt signaling through PKA and PI-3K in neuroectodermal (NE-4C) stem cells [24]. There is a possible association between prostaglandin E2 (PGE2), cyclooxygenase-2 (COX-2), and microsomal prostaglandin E synthase-1 (mPGES-1) [25].

PGE2 treatment causes changes in cell behavior, such as an increase in components of cell motility and proliferation, as well as the expression of Wnt target genes in Wnt-activated cell line NE-4C stem cells in human colorectal adenoma and carcinoma cells [24]. PGE2 increases the final distance and path length traveled in the migration of Wnt-activated neuroectodermal stem NE-4C cells and alters the phenotype of Wnt-treated cells [24]. Wnt/β-catenin signaling occurs through a complex, highly regulated pathway that involves the phosphorylation of multiple sites on β-catenin, which may promote its degradation or activation and subsequent nuclear internalization [24]. PGE2 treatment administered to Wnt-activated cells increased the expression of non-phosphorylated (active) β-catenin (Ser33/37/Thr41) protein. PGE2 signaling may modulate GSK3β activity, although this remains to be determined [24]. Our gene expression results show a potential interaction of PGE2 and the canonical Wnt pathway in the nervous system and provide further evidence for a link to ASD [24]. We show that PGE2 interacts with canonical Wnt signaling through the phosphoinositide-3 kinase (PI-3K) pathway or the protein kinase A (PKA) pathway [24]. PKA and PI-3K produce reported behavioral changes in cell motility and proliferation, as well as gene expression. Specifically, we found that inhibiting these PGE2 downstream pathway kinases, PKA and PI-3K, with H89 and Wort, respectively, reduced the effect of PGE2. Increased PGE2 signaling can modify cell migration, proliferation behavior, and gene expression in Wnt-activated NE-4C stem cells [24]. PGE2 can affect Wnt-dependent cell behaviors and gene expression in neuroectodermal stem cells through PKA and PI-3K [24]. Aberrant PGE2 and Wnt signaling have been linked to ASD [24]. These aberrations may be potential mechanisms in the genesis of neurodevelopment disorders like ASD [24].

A proteomic analysis revealed eight signaling pathways that were significantly dysregulated in ASD patients; three of these (transendothelial leukocyte migration, antigen processing and presentation, and graft vs. host disease) were associated with the neuroimmune response [23]. The metabolism of tryptophan related to the neuroimmune response play a potential role in ASD. Integrated proteome and metabolome analysis showed that six signaling pathways were significantly enriched in ASD patients, three of which were correlated with impaired neuroinflammation (glutathione metabolism, metabolism of xenobiotics by cytochrome P450, and transendothelial migration of leukocytes) [23]. PGE2 levels have been associated with the inflammatory response in ASD. Therefore, the neuroimmune response in ASD provides potential biomarkers as targets for early intervention [23].

Many studies have reported that neuroinflammation, which is also observed in neurodegenerative disorders, is implicated in the pathophysiology of ASD [24]. Moreover, the effect of altered microbiota on neurodegeneration and neuroinflammation in the pathological factors of ASD has been studied [24]. The PGE2 level was considerably greater in patients with ASD than in controls [24]. An association between the impairment of long-chain polyunsaturated fatty acid (LCPUFA) metabolism and both proinflammation and oxidative stress in the etiopathology of ASD has been reported [24]. Therefore, omega-3 LCPUFA supplementation ameliorates proinflammation and oxidative stress-related symptoms [24].

ASD patients exhibit significantly higher levels of the measured parameters compared to neurotypical controls, with the exception of EP2 receptors, which showed an opposite trend [24]. Although the measured parameters did not correlate with the severity of social and cognitive dysfunction, PGE2, COX-2, and microsomal prostaglandin E synthase-1 (mPGES-1) were associated with sensory processing dysfunction [25].

The positive correlation between PGE2, COX-2, and mPGES-1 confirms the role of the PGE2 pathway and neuroinflammation in the etiology of ASD, and the possibility of using PGE2, COX-2, and mPGES-1 as biomarkers of autism severity has been reported [24]. PGE2 is an endogenous lipid molecule involved in normal brain development. PGs include PGD2, and PGE2 interacts with G protein-coupled receptors, presenting new opportunities for transformative therapeutics in neuropsychiatric disorders [23]. COX-2- mice express increased hyperactivity, repetitive behavior, and social abnormality [4]. The COX2/PGE2 pathway changes neuronal cell behavior and the expression of ASD-related genes and proteins [22]. Mouse models of ASD are related to disrupted COX2/PGE2 signaling [26]. PGE2 activates the canonical Wnt signaling pathway, and the upregulated genes have been previously reported to be associated with ASD [26]. PGE_2_ signaling also converges with the Wnt canonical pathway in the developing mouse brain after maternal exposure to PGE_2_ at the level of β-catenin or through PKA and PI-3K in neuroectodermal (NE-4C) stem cells [26]. Additionally, prenatal exposure to PGE_2_ affects the Wnt pathway at the level of β-catenin, which is the major downstream regulator of Wnt-dependent gene transcription [23]. Regarding the function of PGE2 and Wnt in the pathophysiology of ASD, PGE2 can affect Wnt-dependent cell activities and gene expression in neuroectodermal stem cells through the interplay of camp/protein kinase A (PKA) and phosphatidylinositol-3-kinase (PI3K) signaling [26]. PGE2 treatment elicits changes in cell activities through increased components of cell motility and proliferation, as well as the expression of Wnt target genes in Wnt-activated NE-4C stem cells [27] (Iwasa). PGE2 increases the final distance and path length traveled, as well as the average speed of migration, in Wnt-activated neuroectodermal stem NE-4C cells [27]. These changes in cell motility and proliferation activity could have critical effects on the early development of the nervous system [27]. The system of cellular events, including migration and proliferation, occurs during developmental windows [27]. Changed migration and proliferation due to irregular gene expression during embryonic development may be potential mechanisms in the genesis of neurodevelopment disorders, such as ASD [27]. These crucial neurobiological processes are required for the formation of complex layered structures in the cerebral cortex, hippocampus, and cerebellum [27].

PGD2 enhances microglial activation and may be involved in neuronal death in the hippocampus [28]. PGE2 may play a role in upstream regulation in the striatum [29]. PGI2 acts as a pathophysiological mediator and therapeutic agent in major inflammatory-mediated disease processes, including rheumatoid arthritis [29].

Neuroinflammation is caused by PG synthase signaling pathways in the lateral cortex of the impacted hemisphere. COX1 and 2 were found in degenerating neurons [29]. COX1 and COX2 seem to cooperatively participate in neurodegeneration during the early period after trauma and are likely to exacerbate neurodegeneration via PG signaling following traumatic brain injury [30].

## 8. PG-Related Signaling Pathways

Wnt signaling modulates neurite outgrowth, axon growth and guidance, dendritic development and arborization, synapse formation, and synaptic plasticity [27]. Moreover, Wnt signaling is crucial in the specification and differentiation of neuronal precursors in the midbrain and forebrain. Wnt signaling is included in a myriad of regulatory processes in the development and organization of the nervous system [27].

Wnt proteins are a key factor in neural tube formation, neuronal migration, and differentiation [26]. Defective Wnt signaling could contribute to the pathogenesis of psychiatric disorders, including ASD [27]. Particularly, Wnt2, located in the putative speech and language region at chromosome 7q31-33, has been identified as a susceptibility gene for ASD [27]. The interaction of the Wnt and PGE2 pathways occurs in NE-4C stem cells: alterations in the levels of PGE2 via endogenous and exogenous means greatly affect nervous system development [27]. Wnt/β-catenin signaling occurs through a complex, highly regulated pathway that involves the phosphorylation of multiple sites on β-catenin, which may promote its degradation or activation and subsequent nuclear internalization [27]. COX-2 is a crucial mediator of inflammation and prostanoid signaling [27]. The efficacy of the COX-2 inhibitor drug, celecoxib, as an adjunctive therapy in the treatment of ASD has been indicated [27]. Mmp-9, a membrane of the matrix metalloproteinase (MMP) family, is also important in neuronal development [27]. Elevated levels of MMP9 protein were found in the amniotic fluid of individuals with ASD [27].

PGE2 is the signaling molecule converted by COX-2. The effect of COX-2-KI on microglial density and morphology may be important in the developing brain [3]. COX-2-deficient (COX-2-KI) mice exhibit sex-dependent molecular changes in the brain and ASD-related behaviors [3]. Microglia may be influenced by the surroundings, affecting the development of the healthy brain with synaptogenesis, synaptic pruning, and phagocytosis. COX-2-KI female rats were affected on gestational day 19 (G19), showing increased microglial density, altered percentages of amoeboid and ramified microglia, changes in branch length (reduced or increased), and decreased branching networks in a region-specific manner [3]. G19 COX-2-KI male rats showed increased microglial density, a higher percentage of ramified microglia, and increased branch counts [3]. Therefore, COX-2 deficiency in the ASD mouse model affects microglia morphology in a sex-, region-, and stage-dependent manner.

As an endogenous lipid molecule, PGE2 is essential for normal brain development. Autism-related behaviors were prominent in male COX-2- mice in most behavioral tests [24]. The involvement of disrupted COX2/PGE2 signaling in ASD pathology has been reported with age-related differences and a greater impact on men. COX-2- mice exhibited increased hyperactivity, repetitive behavior, and social abnormality [24]. The COX2/PGE2 pathway changed neuronal cell behavior and the expression of ASD-related genes and proteins. Mouse models have shown disrupted COX2/PGE2 signaling [24]. Aberrant cell migration and proliferation are pathophysiologic mechanisms affecting the brain and may contribute to the occurrence of neurodevelopment disorders [26].

Disrupted COX2/PGE2 signaling is involved in ASD pathology with age-related differences and a greater impact on males. We propose that (COX)-2- mice might serve as a novel model system to study specific types of autism [31]. The COX/PGE pathway plays an important role in synaptic plasticity and may be involved in the pathophysiology of ASD. Prenatal exposure to PGE_2_ affects the Wnt pathway at the level of β-catenin, the major downstream regulator of Wnt-dependent gene transcription [31]. In this regard, plasma transferrin, which is an iron mediator related to eicosanoid signaling, may be related to the social impairment seen in individuals with ASD [32]. Increasing endocannabinoid 2-arachidonoylglycerol (2-AG) may cause neuroinflammation-associated Aβ42 accumulation and neurodegeneration [33]. Prostaglandins (PG) and cytokines have been discovered to play a major role in inflammatory processes. The balance between pro- and anti-inflammatory compounds is nowadays a challenge for more targeted therapeutic approaches [34]. According to a recent study on urine proteomic and metabolomic profiling, PGE2, which is an ARA metabolite of COX signaling, is an endogenous lipid molecule and plays a key role in the early development of the nervous system [33]. The COX-2 deficiency in the aforementioned ASD mouse model influences microglia morphology in a sex-, region-, and stage-dependent manner [34].

Many studies have reported that the risk of ASD increases after prenatal exposure to environmental pesticides. Environmental factors, such as exposure to drugs, toxins, or infectious agents, cause disruptions in PGE2 signaling by increasing oxidative stress levels, consequent lipid peroxidation, and the immunological response, resulting in abnormalities in PGE2 signaling and inducing symptoms similar to those of ASD [23]. Abnormal PGE2 levels, due to environmental insults during prenatal development, have been linked to brain pathologies because of destabilization of the actin cytoskeleton at various stages of neuronal differentiation via the dissociation of unbound PKA-phosphorylated ser94-spinophilin, resulting in changes in neuronal morphology [35]. Prenatal exposure to single chemicals, such as the per- and polyfluoroalkyl substances, is associated with biological perturbations in the mother, fetus, and placenta, as well as with adverse health outcomes related to PGs [36]. Dysregulation of the oxidative stress response and proinflammatory processes are the main etiological geneses of ASD pathogenesis [32]. Moreover, altered migration and proliferation due to irregular gene expression during embryonic development in ASD have been previously reported [32]. Furthermore, phospholipid-related metabolites and lipid transporters and mediators are impaired in different pathological conditions in the ASD etiology [32].

AChE inhibition, disruption of voltage-gated sodium channels, and GABA inhibition have been investigated in this context [37]. Prenatal chlorpyrifos (CPF) is a widely diffused environmental toxicant associated with neurobehavioral deficits and increased risk of ASD occurrence in children due to the considerable influence of oxidative stress [38]. Organophosphate insecticides, including CPF, are widely diffused environmental toxicants associated with neurobehavioral deficits and increased risk of ASD occurrence in children [38]. Oxidative stress and dysregulated immune responses are implicated in both organophosphates’ neurodevelopmental effects and ASD etiopathogenesis. At birth, Black and tan brachyury T+tf/J (BTBR) mice exhibited greater oxidative stress processes [38]. The increased oxidative stress levels during their early postnatal life could result in delayed and long-lasting alterations in specific pathways relevant to ASD, of which PGE2 signaling represents an important one [38]. Since microglia normally transition between an amoeboid or ramified morphology depending on their surroundings and are important for the development of the healthy brain, assisting with synaptogenesis, synaptic pruning, and phagocytosis, the COX-2 deficiency in our ASD mouse model is considered to influence the microglia morphology in a sex-, region-, and stage-dependent manner [3].

## 9. PG-Related Neuroinflammation

Neuroinflammation, which is characterized by dysregulated inflammatory responses in the brain and spinal cord, has a pivotal role in the onset of several neurodegenerative disorders [39]. PG plays a major role in inflammatory processes [40]. PGs and cytokines are abundantly produced in the body and are considered crucial mediators of the immune response. These discoveries have opened up a new area and concept of inflammation as they have provided an overview of PG and cytokine implication in various diseases [40]. The resulting acetylated COX2 induces specialized pro-resolving mediators (SPMs) by neurons related to microglial activity. The indirect monitoring of microglia by neuronal Sphk1 may provide anti-inflammatory functions, which are investigated in other neurodegenerative disorders [35].

Sphingolipids, a lipid class characterized by a long-chain amino alcohol backbone, their metabolic intermediates, and enzyme systems, such as COXs, sphingosine kinases, and sphingomyelinase, have interconnected signaling pathways in the CNS [40]. Researchers have studied the major enzymes involved in sphingolipid metabolism, novel metabolic intermediates such as N-acetyl sphingosine, their complex interactions in CNS physiology, the disruption of their function in neurodegenerative disorders, and therapeutic strategies targeting sphingolipids for improved drug development [40]. Sphingolipids are involved in neuroinflammation and neurodegeneration [40]. The resulting acetylated COX2 facilitates the synthesis of the stereochemistry of specialized pro-resolving mediators (SPMs), which target the metabololipidomics of resolution–inflammation metabolomes [41]. The neuron then acts on the microglia, enhancing the phagocytosis of the microglia. Sphk1 is an indirect regulator of microglial phagocytic activity that involves neuronal actions [42]. Additionally, regarding PG and inflammation, myeloid cell bioenergetics are suppressed in response to increased signaling by the lipid messenger PGE_2_, a major modulator of inflammation in aging mice [42]. In aging macrophages and microglia, PGE2 signaling through its EP2 receptor facilitates the sequestration of glucose into glycogen, thereby reducing glucose flux and mitochondrial respiration. This energy-deficient state, which drives maladaptive proinflammatory responses, is further augmented by a dependence of aged myeloid cells on glucose as a principal fuel source [41]. Inflammation is a ubiquitous factor accompanying neurodegeneration, and a recent study has reported the major contribution of inducible COX-2 and its downstream PG signaling pathways in modulating the neuroinflammatory responses and neuronal function [42].

Regarding the mechanisms of inflammation, microglial activation results in the overproduction of inflammatory mediators in the pathogenesis of neurodegeneration and neurotoxicity [43]. Many phytochemicals exhibit decreased microglial activation and production of inflammatory mediators [43]. Phytochemicals are possible candidates for maintaining normal CNS homeostasis by counteracting neuroinflammation and neurodegeneration [43].

The activation of microglia releases various proinflammatory cytokines and ROS, resulting in neuronal cell degeneration [43]. The anti-neuroinflammatory properties of lipopolysaccharide, proinflammatory cytokines, and growth factors induce COX-2 expression, and this activation controls PGE2 synthesis [43]. The release of proinflammatory cytokines [tumor necrosis factor (TNF), IL-1, and IL-6] from the microglia is increased during neuroinflammation [43]. Chloroform- and hexane-soluble fractions of *A. holophylla* and their active compounds greatly inhibited LPS-induced ROS and NO production with increased COX-2 and PGE2 expression [43]. Mitogen-activated protein kinase (MAPK) is the key signaling mechanism responsible for controlling inflammatory pathways in the LPS-activated microglia. The activation of the MAPK signaling pathway induces the release of proinflammatory cytokines, including IL-6, IL-1, and TNF [43]. The inhibition of *N*-terminal kinase (JNK) signaling suppresses the secretion of IL6, IL-1, and TNF, inhibiting inflammation. The p38 (MAPK) pathway is responsible for regulating proinflammatory cytokines and plays an important role in suppressing inflammation [43]. Terpenoids, which were produced from the chloroform and hexane fractions, showed similar anti-neuroinflammatory effects with considerable inhibition of nitric oxide and ROS production. These terpenoids inhibited the phosphorylation of c-Jun JNK, which further inhibited the production of proinflammatory mediators, including PGE2 and interleukins (IL-6 and IL-1β) [43]. Therefore, the chloroform- and hexane-soluble fractions mediated MAPK inhibition and the JNK pathway, lowering the inflammatory cascades in the microglia [43].

The endocannabinoid system plays an important role in balancing the inflammatory and redox status and synaptic plasticity, as a potential target for ASD pathophysiology [44]. Of reference, increased endocannabinoid 2-arachidonoylglycerol (2-AG) by inhibiting the action of monoacylglycerol lipase (MAGL) prevented PGE2 production, neuroinflammation-associated Aβ42 accumulation, and neurodegeneration, indicating that it is a good therapeutic target for relieving cognitive impairment [45].

## 10. Urinary Metabolites in the Pathophysiology of ASD

The abnormal metabolic pathways in ASD children present in their urine and blood. Potential common pathways shared by animal and human studies related to the improvement of ASD symptoms after pharmacological interventions were mammalian–microbial co-metabolites, purine metabolism, and fatty acid oxidation [34].

Significant urinary metabolites, including prostaglandin E2, phosphonic acid, lysine, threonine, and phenylalanine, exhibit associations with ASD. Additionally, the involvement of the phosphatidylinositol and inositol phosphate pathways may be included in the pathophysiology of ASD [46]. The potential role of PGE2-EA in autistic children was observed to be reduced compared to typically developing children. PGE2-EA is a naturally occurring neutral lipid metabolite of PGs synthesized in vivo through COX-facilitated oxygenation of arachidonoyl ethanolamine [47]. Lower PGE2-EA levels have been observed in the urine of autistic children compared to healthy controls [46]. Alterations of the phosphatidylinositol and inositol phosphate pathways may contribute to the pathophysiology of ASD [47].

## 11. Signaling Mediator

PG signaling governs various physiological and pathological processes, including cardiovascular homeostasis and inflammation. During development, PGE2 signaling regulates embryogenesis, hepatocyte differentiation, hematopoiesis, and kidney formation [48].

Abnormalities in the PGE2 signaling pathway have been associated with neurodevelopmental disorders, such as ASD, via an increased amplitude of calcium fluctuation in the neuronal growth cones, affecting the length of neurite extension. Therefore, PGE2 may affect intracellular calcium dynamics in differentiated neuronal cells and possibly affect the early development of the nervous system [48]. PGE2 signaling has been reported as an important correlating factor between oxidative stress and ASD [49].

PGJ2 and its metabolites have a cyclopentenone ring with reactive α,β-unsaturated carbonyl groups that form covalent Michael adducts with key cysteines in proteins and GSH. Cysteine-binding electrophiles such as PGJ2 are considered to play an important role in determining whether neurons will live or die [47].

## 12. Metabolism

PGs are synthesized from ARA through the catalytic activities of COX, whereas the production of different PG types, such as PG F2 alpha (PGF) and PG E2 (PGE), is regulated by specific PG synthases (PGFS and PGES) [47]. PGF2a and TXA2 are endogenous ARA metabolites, influencing diverse physiological processes such as inflammation by activating the PGF_2α_ receptor (FP) and TXA_2_ receptor (TP) [50]. These reveal structural features not only for the specific recognition of endogenous ligands and the attainment of receptor selectivity of FP and TP but also the common mechanisms of receptor activation and Gq protein coupling, such as the PGD2 receptor (DP1-2), PGE2 receptor (EP1-4), FP, PGI2 receptor, and TP [50].

Considerable data have indicated that COX enzymes and ARA metabolites play important roles in the initiation and development of human diseases [12]. ARA is biotransformed via various routes into several mediators in the regulation of inflammatory processes [51]. Prostaglandin E2 may inhibit the production of inflammatory cytokines, causing inflammatory events such as mast inflammatory leukotriene production [52]. Based on these conflicting experimental data, ARA may be a key mediator of neuroinflammation [52].

Polyunsaturated fatty acid interventions induce microglial activation, altering cell morphology. The intricate nature of fatty acid interactions with microglial cells and their potential implications for neuroinflammation have also been previously reported [39].

The ARA pathway is associated with many inflammatory diseases. Esterified ARA on the inner surface of the cell membrane is hydrolyzed to its free form by PLA2, which is in turn further metabolized by COXs, LOXs, and CYP enzymes to a spectrum of bioactive mediators that includes prostanoids, leukotrienes (LTs), epoxyeicosatrienoic acids (EETs), dihydroxyeicosatetraenoic acid (diHETE), eicosatetraenoic acids (ETEs), and lipoxins (LXs). Considerable data indicate that COX enzymes and their metabolites of ARA play important roles in the initiation and development of human diseases [53].

The potential role of PGE2-EA in children with ASD was found to be reduced compared to typically developing children. PGE2-EA is derived from a neutral lipid metabolite of synthesized PGs and may exhibit actions similar to those of PGE2. PGE2-EA is involved in the inflammatory response of human monocyte lineage cells [51]. Alterations to the phosphatidylinositol and inositol phosphate pathways may contribute to the pathophysiology of autism. Significant urinary metabolites, including PGE2, phosphonic acid, lysine, threonine, and phenylalanine, are reportedly associated with autism. Therefore, the involvement of the phosphatidylinositol and inositol phosphate pathways suggests their potential role in the pathophysiology of ASD [54].

## 13. Conclusions

PGI2 is synthesized from ARA by metabolic-pathway-dependent COX and synthesized in the primary step of ARA transformation (PGG2, PGH2) by the degradation of the abovementioned PGs, or are not presented physiologically and may be chemically generated (PGK2, PGL2). PGG_2_ and PGH_2_ are produced from ARA by the COX and peroxidase activities of the enzymes. COX-2 is induced in response to inflammatory and mitogenic stimuli and generates PGE_2_ and PGI_2_ (prostacyclin) in inflammatory cells.

PGs are a group of endogenously produced compounds modulating the neurophysiological function of the human body. PGs are synthesized from ARA by the action of COXs and terminal PG synthases [2]. Esterified ARA on the inner surface of the cell membrane is hydrolyzed to its free form by PLA2, which is in turn further metabolized by COXs and lipoxygenases (LOXs) and cytochrome P450 (CYP) enzymes to a spectrum of bioactive mediators that includes prostanoids, leukotrienes (LTs), epoxyeicosatrienoic acids (EETs), dihydroxyeicosatetraenoic acid (diHETE), eicosatetraenoic acids (ETEs), and lipoxins (LXs). Considerable data have indicated that COX enzymes and ARA metabolites play important roles in the initiation and development of human diseases [51].

Endogenous PGs produce detrimental effects due to their proinflammatory properties. PGs consist of PGD_2_ and PGE_2_, through G protein-coupled receptors. PGG_2_ and PGH_2_ are produced by COX isozymes, such as COX-1 and COX-2. PGE2 levels are considerably elevated in individuals with ASD compared to controls. The significant positive correlations reported between 8-isoprostan, leukotriene, and PGE2 confirmed the association between the impairment of LCPUFA metabolism and proinflammation and oxidative stress in the etiopathology of ASD. Disrupted COX2/PGE2 signaling is also involved in ASD pathology with age-related differences and a greater impact on men. COX-2- mice showed increased hyperactivity, repetitive behavior, and social abnormality. The COX2/PGE2 pathway changes neuronal cell behavior and the expression of ASD-related genes and proteins. Glutamate excitotoxicity was proposed to participate in the propagation of ASD. PGE2 is a physiological signal for inducing glutamate release from the subventricular zone in astrocytes. Therefore, PEG 2E may be closely related to glutamate toxication, contributing to the development of ASD. The convergence of PGE2 with the Wnt canonical pathway was apparent due to a higher level of PKA-activated phosphorylated β-catenin (Ser552) and active non-phospho β-catenin (Ser33/37/Thr41). Abnormal PGE2 levels in the mouse brain during development may interfere with the Wnt pathway via phosphorylation of β-catenin at multiple sites, which can lead to differential expression of crucial neurodevelopmental genes.

## Figures and Tables

**Figure 1 cimb-47-00071-f001:**
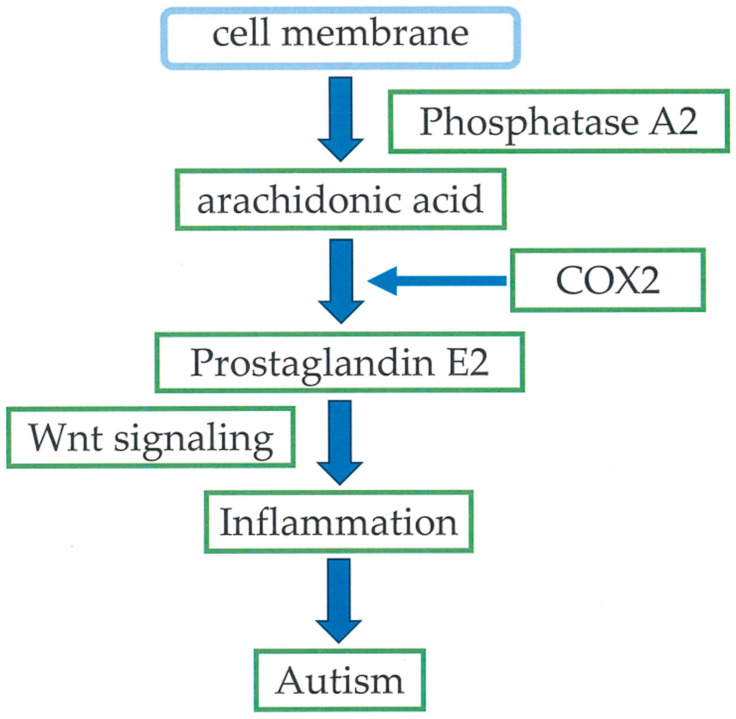
Metabolisms of prostaglandins.

**Figure 2 cimb-47-00071-f002:**
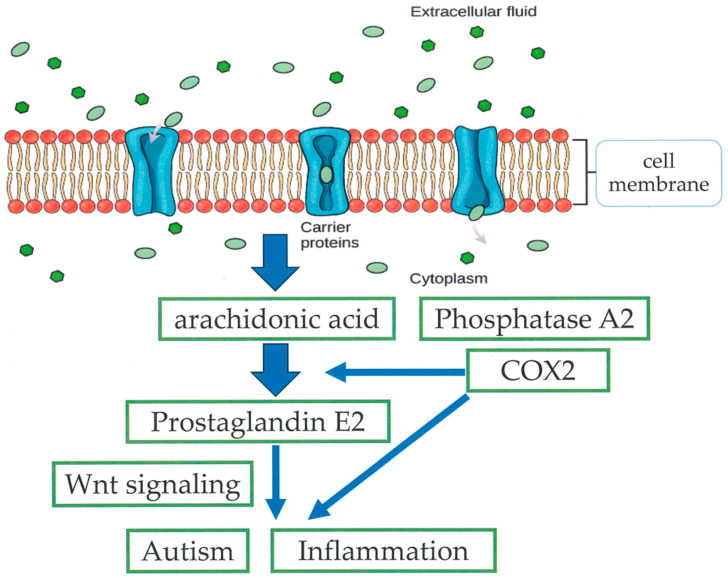
The associations among prostaglandin, oxidative stress, and cyclooxygenase (COX2).

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
