# Peer review of "Prostaglandins: Biological Action, Therapeutic Aspects, and Pathophysiology of Autism Spectrum Disorders"

_cimb, 2025, doi:10.3390/cimb47020071_

Round 1
Reviewer 1 Report (Previous Reviewer 3)
Comments and Suggestions for Authors
Some previous comments are still to be further elucidated, i.e.:
Overview: the relationship between Pas and ASD remains elusive and only speculative. More should be added on ASD biology and possible involvement of PGs. Several parts are confusing also by mistakes in editing.
If the authors want to discuss the relationship between PG and ASD should at least introduce ASD too.
Author Response
Reviewer 1 comment(Round 1)
Open Review
(x) I would not like to sign my review report
( ) I would like to sign my review report
Quality of English Language
(x) The quality of English does not limit my understanding of the research.
( ) The English could be improved to more clearly express the research.
Comments and Suggestions for Authors
Some previous comments are still to be further elucidated, i.e.:
Overview: the relationship between Pas and ASD remains elusive and only speculative. More should be added on ASD biology and possible involvement of PGs. Several parts are confusing also by mistakes in editing.
If the authors want to discuss the relationship between PG and ASD should at least introduce ASD too.
Our answer
We discussed the relationship between PG and ASD marked as sky blue (sections 1, 4, 6, 7)
Submission Date
07 November 2024
Date of this review
08 Nov 2024 10:06:13
Reviewer 2 Report (New Reviewer)
Comments and Suggestions for Authors
Esterified arachidonic acid (ARA) on the inner surface of the cell membrane is hydrolyzed to its free form by phospholipase A2 (PLA2), which is further metabolized by COXs and lipoxygenases (LOXs) and cytochrome P450 (CYP) enzymes. PGs produce detrimental effects due to their pro-inflammatory properties. The generation of prostaglandin (PG)G2 and PGH2 was produced by cyclooxygenase (COX) isozymes such as COX-1 and COX-2. Prostaglandin E2 (PGE2) was significantly elevated levels in ASD. Considerable data indicate that COX enzymes, and their metabolites of ARA play important roles in the initiation and development of human neurodevelopmental diseases. The involvement of disrupted COX2/PGE2 signaling in ASD pathology in change neuronal cell behavior and expression of ASD-related genes and proteins due to disrupted COX2/PGE2 signaling.
A well written timely manuscript addressing an unmet need in the field. Multiple strengths.
Minor suggestion to strengthen this paper.
Line 341, “This suggests that 2-AG could serve as a promising therapeutic target for alleviating cognitive deficits [45].”
1) Suggest adding on a subheading for the endocannabinoid system.
Including the fact that Palmitoylethanolamide (PEA) is an endocannabinoid proven to prevent glutamatergic toxicity and inhibit inflammatory responses simultaneously [14], which authors rightfully inserted lines 77-78.
2) Suggest expanding the above short paragraph (lines 336-342) and include that multiple lines of evidence suggest a central role for the endocannabinoid system (ECS), which comprises receptors, ligands, and associated enzymes, in the neurocognitive development and function and in the pathogenesis of fragile X syndrome (FXS) (reviewed in Palumbo et al., 2023 https://doi.org/10.1186/s11689-023-09475-z).
FXS is caused by a full-mutation in the fragile X messenger ribonucleoprotein 1 (FMR1) gene, which results in reduced or negligible levels of FMR1 protein (FMRP), and is the most common monogenetic cause of autism spectrum disorder (ASD) (https://doi.org/10.1159/000330213).
Enzymes that function in synthesizing 2-AG, which is released from postsynaptic terminals, include phospholipase C and diacylglycerol lipase (DAGL). The loss of FMRP function is thought to cause downregulated EC in FXS via altered DAGL function and EC excitatory and inhibitory signaling in neuronal synapses, with downstream dysregulation of the EC signaling in the CNS that may contribute to the clinical abnormalities observed in FXS, including clinical features of ASD in FXS (https://doi.org/10.1542/peds.2016-1159F; Budimirovic, D. B., et al. "Consensus of the fragile X clinical and research consortium on clinical practices autism spectrum disorder in fragile X syndrome." Walnut Creek, California: National Fragile X Foundation Website (2014).
Together, published placebo-phase clinical trials data on use of synthetic Cannabidiol (CBD) in patients with FXS (Berry-Kravis et al., 2022 https://doi.org/10.1186/s11689-022-09466-6) and incoming open-label data on the large cohort of patients with FXs that continue on CBD suggests the tangible potential role of CBD as a treatment for FXS.
Note: Above references are examples of key references that as such should be considered to be cited as they do contribute to the relevance of the above stated, but they are not the only ones for some aspects of the above paragraph.
3) typos in the body text needs to be corrected. For example
6. Prostaglandin signaling in pathophysiologyof ASD
6.2 the matrilysin also knownactivity...
4) This suggestion should not impact that this paper deserve to be published, but future readers would certainly benefit if authors could additionally polish and organize sections 7-9, and 12 (conclusions), given that they are densely written. The additional work would make this valuable info easier to follow and 'digest.'
Comments on the Quality of English LanguageNo major suggestions.
Author Response
Reviewer 2 comment(Round 1)
Open Review
( ) I would not like to sign my review report
(x) I would like to sign my review report
Quality of English Language
( ) The quality of English does not limit my understanding of the research.
(x) The English could be improved to more clearly express the research.
Comments and Suggestions for Authors
Esterified arachidonic acid (ARA) on the inner surface of the cell membrane is hydrolyzed to its free form by phospholipase A2 (PLA2), which is further metabolized by COXs and lipoxygenases (LOXs) and cytochrome P450 (CYP) enzymes. PGs produce detrimental effects due to their pro-inflammatory properties. The generation of prostaglandin (PG)G2 and PGH2 was produced by cyclooxygenase (COX) isozymes such as COX-1 and COX-2. Prostaglandin E2 (PGE2) was significantly elevated levels in ASD. Considerable data indicate that COX enzymes, and their metabolites of ARA play important roles in the initiation and development of human neurodevelopmental diseases. The involvement of disrupted COX2/PGE2 signaling in ASD pathology in change neuronal cell behavior and expression of ASD-related genes and proteins due to disrupted COX2/PGE2 signaling.
A well written timely manuscript addressing an unmet need in the field. Multiple strengths.
Minor suggestion to strengthen this paper
Our answer
Thank you so mich
Line 341, “This suggests that 2-AG could serve as a promising therapeutic target for alleviating cognitive deficits [45].”
Our answer
We added the sentence such as “This suggests that 2-AG could serve as a promising therapeutic target for alleviating cognitive deficits [45].”marked as red color.
1) Suggest adding on a subheading for the endocannabinoid system.
Including the fact that Palmitoylethanolamide (PEA) is an endocannabinoid proven to prevent glutamatergic toxicity and inhibit inflammatory responses simultaneously [14], which authors rightfully inserted lines 77-78.
Our answer
We added the sentence such as “Palmitoylethanolamide (PEA) is an endocannabinoid proven to prevent glutamatergic toxicity and inhibit inflammatory responses simultaneously [14],
2) Suggest expanding the above short paragraph (lines 336-342) and include that multiple lines of evidence suggest a central role for the endocannabinoid system (ECS), which comprises receptors, ligands, and associated enzymes, in the neurocognitive development and function and in the pathogenesis of fragile X syndrome (FXS) (reviewed in Palumbo et al., 2023 Palumbo et al. 2023)
Our answer
We added the sentences in the section 9 as fellows:
Multiple lines of evidence suggest a central role for the endocannabinoid system (ECS), which comprises receptors, ligands, and associated enzymes in the neurocognitive development and function and in the pathogenesis of fragile X syndrome (FXS) (Palumbo et al, 2023)[45]
FXS is caused by a full-mutation in the fragile X messenger ribonucleoprotein 1 (FMR1) gene, which results in reduced or negligible levels of FMR1 protein (FMRP), and is the most common monogenetic cause of autism spectrum disorder (ASD) (https://doi.org/10.1159/000330213).
Our answer
We added the sentences as fellows “FXS is caused by a full-mutation in the fragile X messenger ribonucleoprotein 1 (FMR1) gene, which results in reduced or negligible levels of FMR1 protein (FMRP), and is the most common monogenetic cause of ASD (Palumbo et al, 2023)”[45].
Enzymes that function in synthesizing 2-AG, which is released from postsynaptic terminals, include phospholipase C and diacylglycerol lipase (DAGL). The loss of FMRP function is thought to cause downregulated EC in FXS via altered DAGL function and EC excitatory and inhibitory signaling in neuronal synapses, with downstream dysregulation of the EC signaling in the CNS that may contribute to the clinical abnormalities observed in FXS, including clinical features of ASD in FXS (https://doi.org/10.1542/peds.2016-1159F; Budimirovic, D. B., et al. "Consensus of the fragile X clinical and research consortium on clinical practices autism spectrum disorder in fragile X syndrome." Walnut Creek, California: National Fragile X Foundation Website (2014).
0ur answer
FXS is caused by a full-mutation in the fragile X messenger ribonucleoprotein 1 (FMR1) gene, which results in reduced or negligible levels of FMR1 protein (FMRP), and is the most common monogenetic cause of ASD (Palumbo et al, 2023) [45]. Multiple lines of evidence suggest a central role for the endocannabinoid system (ECS) in the neuronal development and cognitive function and in the pathogenesis of fragile X syndrome. The ECS mediated feedback inhibition and synaptic plasticity are thought to be disrupted in FXS due to dysregulation of enzymes that are integral to the ECS (Amos Wilson et al, 2008) (The National Fragile X Foundation, 2014) .
Together, published placebo-phase clinical trials data on use of synthetic Cannabidiol (CBD) in patients with FXS (Berry-Kravis et al., 2022 https://doi.org/10.1186/s11689-022-09466-6) and incoming open-label data on the large cohort of patients with FXs that continue on CBD suggests the tangible potential role of CBD as a treatment for FXS.
Our answer
We added the sentences on (Berry-Kravis et al., 2022 https://doi.org/10.1186/s11689-022-09466-6) as fellow:
“Fragile X syndrome (FXS) is associated with dysregulated endocannabinoid signaling and may therefore respond to cannabidiol therapy. CONNECT-FX was a double-blind, randomized phase 3 trial assessing efficacy and safety of ZYN002, transdermal cannabidiol gel, for the treatment of behavioral symptoms in children and adolescents with FXS. CONNECT-FX, ZYN002 was well tolerated in patients with FXS and demonstrated evidence of efficacy with a favorable benefit risk relationship in patients with ≥ 90% methylation of the FMR1 gene, in whom gene silencing is most likely, and the impact of FXS is typically most severe (Berry-Kravis et al.2022[54]. The ZYN002 was tolerated well in patients with FXS and indicated evidence of efficacy with a favorable benefit risk relationship in patients with ≥ 90% methylation of the promoter region of the FMR1 gene with little or no FMRP production, and the impact of FXS is typically most severe (Berry-Kravis et al.2022.”[54]
Note: Above references are examples of key references that as such should be considered to be cited as they do contribute to the relevance of the above stated, but they are not the only ones for some aspects of the above paragraph.
3) typos in the body text needs to be corrected. For example
- Prostaglandin signaling in pathophysiologyof ASD
6.2 the matrilysin also knownactivity...
4) This suggestion should not impact that this paper deserve to be published, but future readers would certainly benefit if authors could additionally polish and organize sections 7-9, and 12 (conclusions), given that they are densely written. The additional work would make this valuable info easier to follow and 'digest.'
Comments on the Quality of English Language
No major suggestions.
Submission Date
07 November 2024
Date of this review
12 Nov 2024 21:10:44
Round 2
Reviewer 1 Report (Previous Reviewer 3)
Comments and Suggestions for Authors
Some major issues still remain.
It is not clear the links to ASD and PGs. Authors should describe at least the ASD molecular pathways involved. In the current form, after a detailed description of PGs, suddenly it should be linked to the spectrum.
What is the significance of microbiome, ASD, and PGs here ?
Comments on the Quality of English LanguageEnglish language requires edits.
Author Response
Comments and Suggestions for Authors
Some major issues still remain.
It is not clear the links to ASD and PGs. Authors should describe at least the ASD molecular pathways involved. In the current form, after a detailed description of PGs, suddenly it should be linked to the spectrum.
What is the significance of microbiome, ASD, and PGs here ?
Our Amnswer
We have added the sentences on the significance of microbiome, ASD, and PGs in this review as mrked green colors as fellow “In summary, the microbiota–gut–brain axis help to understand the pathopyology of ASD and possible treatment for ASD. Microbial fermentation of plant-based fiber can affecte a beneficial effect on development of ASD. The gut microbiota–brain axis help to understand the mechanism in relation to PGE2, and my find possible treatments for ASD”. In the added section 13
Best regards,
Dr. Yui, Dr. Imataka
Round 3
Reviewer 1 Report (Previous Reviewer 3)
Comments and Suggestions for Authors
Perhaps, I was not clear in my comments, or the authors did not understand them.
I was meaning that if you cite glutamate or GABA, as examples, of molecules potentially controlled or regulated by PGEs, and there could be a link with autism, you should describe more in detail how these molecules are dysregulated in ASD, providing more evidences. If not, it only remains speculative.
Author Response
【Response to Reviewer3】
Open Review
Comments and Suggestions for Authors
Perhaps, I was not clear in my comments, or the authors did not understand them.
I was meaning that if you cite glutamate or GABA, as examples, of molecules potentially controlled or regulated by PGEs, and there could be a link with autism, you should describe more in detail how these molecules are dysregulated in ASD, providing more evidences. If not, it only remains speculative.
Submission Date
07 November 2024
Date of this review
10 Dec 2024 09:38:22
Our Answer
We added the association between sentences on the OEGs on the section such as 5. Association of Prostaglandins to other amino acids marker by green colors.

Round 4
Reviewer 1 Report (Previous Reviewer 3)
Comments and Suggestions for Authors
Authors replied : "We added the association between sentences on the OEGs on the section such as 5. Association of Prostaglandins to other amino acids marker by green colors."
However, there is no association explained between GABA or other PEG-regulated molecules and ASD.
Authors failed to answer to my simply question.
Author Response
Comments and Suggestions for Authors
Authors replied:
"We added the association between sentences on the OEGs on the section such as 5. Association of Prostaglandins to other amino acids marker by green colors."
However, there is no association explained between GABA or other PEG-regulated molecules and ASD.
Authors failed to answer to my simply question.
Oue answer:
There are a few fundings on the association between GABA activity. We added the sentences on the association explained between GABA or other PEG-regulated molecules and ASD in the sections 4 and 5 and conclusion marked by green colors.

Round 5
Reviewer 1 Report (Previous Reviewer 3)
Comments and Suggestions for Authors I meant that if the authors want to link the PEG2 pathways with autism, they need to describe more in details that the pathway is indeed involved in autism. As example, it isn't enough to describe the PEG2/GABA (that the authors made in very clear and readable manner), but the question is: is GABA dysregulated in autism ? Authors should provide more on this aspect, of course for all the PEG2-mediated molecular pathways that could affect autism.
Author Response
Comments and Suggestions for Authors
Authors replied:
"We added the association between sentences on the PEGs on the section such as 5. Association of Prostaglandins to other amino acids marker by green colors."
However, there is no association explained between GABA or other PEG-regulated molecules and ASD.
Authors failed to answer to my simply question.
Our answer
We added the sentences on the association explained between GABA or other PEG-regulated molecules and ASD.
Dear Professor Yui,
I hope this email finds you well. Reviewer 1 gave the reject report in the fourth round. We asked him about the specific reasons, and the following is his reply. Please give a specific reply or rebuttal to the reviewer's comments and send it to me via email. " I meant that if the authors want to link the PEG2 pathways with autism, they need to describe more in details that the pathway is indeed involved in autism. As example, it isn't enough to describe the PEG2/GABA (that the authors made in very clear and readable manner), but the question is: is GABA dysregulated in autism ? Authors should provide more on this aspect, of course for all the PEG2-mediated molecular pathways that could affect autism. Yes, the authors could further revise their manuscript before acceptance. " We look forward to hearing from you.
Our answer
We further added the responses to your comment: “the question is GABA dysregulated in autism ?” in the section 5; we inducted the paper such as “Zhao H, Mao X, Zhu C, Zou X, Peng F, Yang W, Li B, Li G, Ge T, Cui R. GABAergic System Dysfunction in Autism Spectrum Disorders. Front Cell Dev. Biol. 2022, 9, 781327. doi: 10.3389/fcell.2021.781327” and described the GABA dysregulated in autism marked by pink color.

Round 6
Reviewer 1 Report (Previous Reviewer 3)
Comments and Suggestions for Authors
no other comments
This manuscript is a resubmission of an earlier submission. The following is a list of the peer review reports and author responses from that submission.
Round 1
Reviewer 1 Report
Comments and Suggestions for Authors
Manuscript ID: cimb-3223866
In this Manuscript, the authors try to review the role of prostaglandins in the pathophysiology of autism spectrum disorders. Although the topic is interesting and could help to find a target to treat autism through the correction of fatty acid metabolism, known to be relate to oxidative stress, neuroinflammation, impaired gut microbiota, and most importantly glutamate excitotoxicity.
But I did not enjoy reading the manuscript. I think that it could be more informative if the flow of text and subtitles were rewritten in a more scientific way.
I have the following comments:
- - Authors did not use the template formatting for review articles correctly (e.g. Material and Methods)
- - Keywords are missing
- - The titles and subtitles were poorly selected and their appearance along the manuscript is not appropriate
- -Legends for figures were missing. And I am sure that these figures have copyright
- -The quality of writing is not good and more efforts are needed to highlight the role of PGs in different etiological mechanisms such as glutamate excitotoxicity, and how it could be related to different co-morbidities in ASD. Role of disrupted COX-2/PGE2 could be easily related to abnormal glutamate signaling.
For example, the authors mentioned that:
“PGE2 is a major ARA metabolite, preferentially formed during the enzymatic activity of COX-2, which localizes to the excitatory glutamatergic neurons and is dependent on the synaptic activity [4].
And the reference is not relevant to this statement
See reference 4:
Rai-Bhogal R.; Wong, C.; Kissoondoyal, A.; Davidson, J.; Li, H. Crawford DA. Maternal exposure to prostaglandin E2 modifies 410 expressions of Wnt genes in mouse brain - An autism connection. Biochem Biophys Rep. 2018,10;14:43-53. doi: 411 10.1016/j.bbrep.2018.03.012.
Comments on the Quality of English LanguageEnglish editing is needed
Reviewer 2 Report
Comments and Suggestions for Authors
the review is very raw and lacks the novelty
-Keywords were not mentioned
- The Manuscript didn't provide sufficient evidence for the role of PG in autism, only superficial background about PG was provided.
-The manuscript should undergo extensive grammatical checking and English editing.
- There is no sufficient illustrations that discuss the signaling pathways linked to autism.
- I didn't understand the role of subtitle "Material and Methods".
The review requires extensive English editing
Reviewer 3 Report
Comments and Suggestions for Authors
- It is hard to understand if this article is a review or a research paper: in example: "Informed Consent Statement: Written informed consent has been obtained from the patient to publish this paper."
makes non sense.
- Paragraph 4. The sentence: "Many studied have reported that ASD are included in neuroinflammation in neuro-degenerative disorders"does not have sense. ASD needs to be shortly introduced, as well as the microbiota sentence is not needed.
- Lines 190 - 193 are the same of lines 201 - 203, as well as 203 - 206.
- Materials and methods : ???
- Overview: the relationship between Pas and ASD remains elusive and only speculative. More should be added on ASD biology and possible involvement of PGs. Several parts are confusing also by mistakes in editing.
Comments on the Quality of English LanguageEnglish language requires editing, i.e. line 206. Authors should read better their final version of the manuscript.